# Urinary Microbiome Dysbiosis and Immune Dysregulations as Potential Diagnostic Indicators of Bladder Cancer

**DOI:** 10.3390/cancers16020394

**Published:** 2024-01-17

**Authors:** Matthew Uzelac, Ruomin Xin, Tianyi Chen, Daniel John, Wei Tse Li, Mahadevan Rajasekaran, Weg M. Ongkeko

**Affiliations:** 1Department of Surgery, Division of Otolaryngology-Head and Neck Surgery, University of California San Diego, La Jolla, CA 92093, USA; muzelac@ucsd.edu (M.U.); rxin@ucsd.edu (R.X.); cathytianyichen04@gmail.com (T.C.); d1john@ucsd.edu (D.J.); wtl008@ucsd.edu (W.T.L.); 2Research Service, VA San Diego Healthcare System, San Diego, CA 92161, USA; 3School of Medicine, University of California San Francisco, San Francisco, CA 94143, USA; 4Department of Urology, San Diego VA Healthcare System, University of California, San Diego, CA 92161, USA

**Keywords:** bladder cancer, urinary microbiome, biomarker, cancer diagnosis, immune modulation

## Abstract

**Simple Summary:**

Current means of bladder cancer diagnosis are invasive or lack sensitivity. Dysbiosis of the urinary microbiome has been implicated in the development of bladder cancer, though its potential as a diagnostic tool is unknown. This study attempts to characterize bladder cancer-specific dysbiosis of the urinary microbiome to explore its diagnostic potential. Numerous species were observed to be differentially abundant between the urine samples of patients with bladder cancer and healthy individuals. Specific immune modulations were observed with respect to these species, as was the enrichment of select pathways known to be implicated in the progression of bladder cancer. The study suggests that the urinary microbiome may reflect dysregulations of the tumor microenvironment. Further investigation may reveal the potential of the identified species as urinary biomarkers of this disease. In this way, the urinary microbiome may allow for the noninvasive and earlier detection of bladder cancer, regardless of stage or subtype.

**Abstract:**

There are a total of 82,290 new cases and 16,710 deaths estimated for bladder cancer in the United States in 2023. Currently, urine cytology tests are widely used for bladder cancer diagnosis, though they suffer from variable sensitivity, ranging from 45 to 97%. More recently, the microbiome has become increasingly recognized for its role in human diseases, including cancers. This study attempts to characterize urinary microbiome bladder cancer-specific dysbiosis to explore its diagnostic potential. RNA-sequencing data of urine samples from patients with bladder cancer (*n* = 18) and matched controls (*n* = 12) were mapped to bacterial sequences to yield species-level abundance approximations. Urine samples were analyzed at both the population and species level to reveal dysbiosis associated with bladder cancer. A panel of 35 differentially abundant species was discovered, which may be useful as urinary biomarkers for this disease. We further assessed whether these species were of similar significance in a validation dataset (*n* = 81), revealing that the genera *Escherichia*, *Acinetobacter*, and *Enterobacter* were consistently differentially abundant. We discovered distinct patterns of microbial-associated immune modulation in these samples. Several immune pathways were found to be significantly enriched with respect to the abundance of these species, including antigen processing and presentation, cytosolic DNA sensing, and leukocyte transendothelial migration. Differential cytokine activity was similarly observed, suggesting the urinary microbiome’s correlation to immune modulation. The adherens junction and WNT signaling pathways, both implicated in the development and progression of bladder cancer, were also enriched with these species. Our findings indicate that the urinary microbiome may reflect both microbial and immune dysregulations of the tumor microenvironment in bladder cancer. Given the potential biomarker species identified, the urinary microbiome may provide a non-invasive, more sensitive, and more specific diagnostic tool, allowing for the earlier diagnosis of patients with bladder cancer.

## 1. Introduction

In the U.S., 82,290 new cases and 16,710 deaths were reported for bladder cancer in 2023, with a 5-year survival rate of 77.9% [1]. These cancers are known to disproportionately affect older, male populations, with the median age at diagnosis being 73 [1]. Veterans are particularly susceptible to developing these diseases due to an increased risk of exposure to chemicals implicated in their pathogenesis, largely Agent Orange [2,3]. Increased usage of tobacco products, as observed in these populations, is also known to increase these individuals’ risk of developing bladder cancer [4,5].

Bladder cancers are classified as either muscle-invasive or non-muscle-invasive, with treatment options varying considerably between these two groups [6,7]. In the treatment of these patients, clinicians will often select a combination of transurethral resection, adjuvant intravesical therapy, radical cystectomy, and neoadjuvant chemotherapy depending on the subtype [6,7]. However, a delayed diagnosis will result in worse prognoses irrespective of interventions [8]. As such, the early detection of bladder cancer is crucial toward improving the outcome of patients with this disease. Of the current means of diagnosis, cystoscopies are often performed when patients become symptomatic, typically presenting with urinary bleeding [9,10]. In these cases, however, the cancers have likely already reached more advanced stages, making treatment more difficult [8]. Moreover, smaller lesions may be overlooked [11,12]. For the early detection of bladder cancer, numerous biomarker screening tests have been developed and implemented [13]. Urine cytology tests, the BTA stat test, and the NMP22 test are commonly used, though they suffer from poor sensitivities in less advanced cancers [13,14,15,16]. Hence, the development of a means to diagnose bladder cancers regardless of their stages or subtypes would provide significant clinical benefit.

The progression from non-muscle-invasive to muscle-invasive bladder cancer has been heavily associated with epithelial–mesenchymal transition (EMT) [17,18,19]. This transition is characterized by a decrease in cell adhesion, by which epithelial cells gain mesenchymal traits [20]. The disruption of this pathway has been associated with increased cell motility and proliferation rates, ultimately promoting the acquisition of muscle-invasive bladder cancer [17,18,19]. The WNT/β-catenin signaling pathway has also been implicated in this transition [21,22]. Mutations in this pathway have been linked to tumorigenesis and the development of drug resistance by inducing a cancer stem cell phenotype [21]. Numerous genetic factors have been identified for their importance in EMT and WNT/β-catenin signaling, including the dysregulation of select transcriptional factors and the differential activation of associated receptors [22,23]. Epigenetic factors are less explored, though the human microbiome has been shown to indirectly regulate both EMT and WNT/β-catenin signaling [24,25].

The human microbiome is a collection of microorganisms that populate the gastrointestinal system [26]. The microbiome has become increasingly implicated in human diseases, including inflammatory bowel disease, psoriasis, and diabetes, among others [27,28]. The microbiome is thought to influence an array of biological pathways, largely through metabolite-mediated immune modulation [29,30]. As such, studies have also characterized the microbiome for its implication in various cancers, particularly colorectal cancers [31,32,33,34]. Less is known regarding the microbiome’s influence beyond the gastrointestinal system, though microbial species are known to exist outside of the gut. Although previously considered sterile, the bladder has been shown to harbor a diverse population of microbiota, creating potential for the urinary microbiome to influence urological diseases [35]. In fact, studies have shown select genera to be enriched in the urine samples of patients with bladder cancer [36,37]. Urinary tract infections, too, have been shown to increase the risk of bladder cancer mortality [38]. Nonetheless, the exact mechanisms by which the urinary microbiome may mediate bladder cancer remain unknown. We have previously demonstrated the importance of the intratumoral microbiome to EMT in bladder cancer [39]. We hypothesize that by investigating the urinary microbiome for similar dysbiosis, we may identify additional microbial influences of this disease. Further, by exploring this dysbiosis for its implications in immune modulation, EMT, and WNT/β-catenin signaling, we may better understand the human microbiome’s relevance to the transition between bladder cancer subtypes. With the discovery of microbial biomarkers, we may be more equipped for the early diagnosis of bladder cancers, regardless of stage or subtype.

To test our hypothesis, RNA-sequencing data of bladder cancer (*n* = 18) and normal (*n* = 12) urine samples were first downloaded from the NCBI BioProject Database. Sequences were mapped to bacterial sequences to yield species-level abundance approximations. Both population-level and species-level variations were explored, as were microbial-associated immune dysregulations and the microbial-associated enrichment of EMT and WNT/β-catenin signaling. Several species that were differentially abundant between the bladder cancer and normal samples were found to be of similar significance in a validation dataset. By assessing the urinary microbiome for its relation to these factors, we hope to demonstrate the importance of biomarker microbiota in the early diagnosis of bladder cancer.

## 2. Materials and Methods

### 2.1. Data Acquisition

RNA-Seq data were downloaded from the NCBI BioProject Database (https://www.ncbi.nlm.nih.gov/gds (accessed on 25 July 2023)) of bladder cancer (*n* = 18) and normal (*n* = 12) urine samples from three studies (PRJNA657414, PRJNA872870, and PRJNA723026). For validation, RNA-Seq data were similarly downloaded from the Genome Sequencing Archive (https://ngdc.cncb.ac.cn/gsa/ (accessed on 3 January 2024)) of bladder cancer (*n* = 62) and normal (*n* = 19) urine samples (PRJCA003781).

### 2.2. Bacterial Read Mapping

Sequences were mapped to bacterial species using the Pathoscope 2.0 software [40], with reference sequences sourced from the NCBI Nucleotide Database (https://www.ncbi.nlm.nih.gov/nucleotide/ (accessed on 28 July 2022)). Bacterial reads were targeted for quantification, and human reads were filtered. Standard parameters were chosen.

### 2.3. Gene Read Mapping

Sequences were mapped to the hg38 reference genome using the STAR 2.7.10a software [41], with reference sequences sourced from the NCBI Nucleotide Database (https://www.ncbi.nlm.nih.gov/nucleotide/ (accessed on 28 July 2022)). Standard parameters were chosen.

### 2.4. Cross Study Normalization

Median Absolute Deviation (MAD) normalization was performed to increase the validity of cross-study comparisons. In this technique, the median expression value of a gene is subtracted from each individual sample’s expression of that gene. The resultant values are then divided by the MAD of that gene to yield relative expression values in each sample. This technique was similarly performed on species abundance values. MAD normalization assumes an equal median and distribution across samples and is resilient to outliers. MAD normalization has been shown to be robust in gene expression experiments compared to other standard normalization tools [42]. Principle coordinate analysis (PCoA) was conducted to confirm the effectiveness of this technique using Euclidean distance.

### 2.5. Microbial Contamination Correction

There is potential for contaminant species to be introduced upon sample collection and sequencing [43]. Across all samples, noncontaminant species are expected to be of greater abundance in samples of a greater total abundance of taxa. Contaminant species are likely introduced in a fixed amount and will not display this behavior. Spearman’s correlations were computed between each species and the total abundance of all species in a sample. Species that did not show a significant correlation to the total abundance of taxa were deemed contaminants and excluded from subsequent analyses.

### 2.6. Global Indicator Analyses

The microbiome R package was used to perform PCoA using Euclidean distance. This package was also used to calculate alpha and beta diversity values.

### 2.7. Differential Abundance Analyses

The Kruskal–Wallis test was used to identify species differentially abundant between bladder cancer and normal samples (*p* < 0.05). A hypergeometric test was used to determine whether a significant number of genera were common to both the original and validation datasets.

### 2.8. Gene Set Enrichment Analyses

The clusterProfile R package was used to assess immune, adherens junction, and WNT signaling pathway enrichment [44]. Twenty-two immune-associated gene sets were sourced from the KEGG PATHWAY Database (https://www.genome.jp/kegg/pathway.html (accessed on 3 August 2023)). The enrichplot R package was used to visualize enrichment results.

### 2.9. Expression Correlation Analyses

Spearman’s correlations were calculated relating species’ abundances to cytokine expression. This analysis was similarly performed to relate species’ abundances to the expression of the genes of the adherens junction and WNT signaling pathways. Only differentially abundant species were analyzed.

## 3. Results

### 3.1. Cross-Study Normalization and Contamination Correction

Bacterial abundance and gene expression counts were extracted from bladder cancer (*n* = 18) and normal (*n* = 12) urine samples spanning three studies (Figure 1A). MAD normalization was performed to increase the comparability of samples for subsequent analyses (see Section 2). PCoA was conducted to visualize the samples’ bacterial abundance and gene expression profiles by study both before (Figure 1B) and after (Figure 1C) MAD normalization. After normalization, the samples showed closer proximity, and thus greater similarity in their expression and abundance profiles, confirming the effectiveness of the chosen normalization technique and demonstrating the comparability of the samples’ abundance and expression profiles for subsequent analyses.

To identify and exclude potential contaminant species, Spearman’s correlations were computed between each species and the total abundance of all species in a sample (see Section 2). Species that did not show a significant correlation to the total abundance of taxa were deemed contaminants and excluded from subsequent analyses. This procedure was performed on bladder cancer and normal samples collectively. A phylogenic tree was created to visualize the distribution of contaminant species by class or phylum (Figure 1D).

### 3.2. Global Urinary Microbiome Dysbiosis

PCoA was conducted to characterize the samples’ bacterial abundance profiles by their disease states (Figure 2A). This analysis attempts to simplify the abundance values of all microbial features in each sample into several arbitrary dimensions. In these dimensions, the proximity of one sample to another describes wholistically the samples’ similarities in microbiome composition. Bladder cancer samples occupy a distinct region of the plot from normal samples, suggesting the presence of a wholistic variation in urinary microbiome composition between these disease states.

Measures of alpha and beta diversity were further computed in each sample (Figure 2B). Absolute and relative diversity were found to be significantly lesser in bladder cancer samples, while Shannon diversity was significantly greater (Figure 2C). It is unclear what factors may have influenced the observed difference in direction. Nonetheless, these metrics suggest the presence of global dysbiosis in the urinary microbiome of bladder cancer patients.

### 3.3. Differentially Abundant Species

The Kruskal–Wallis test was used to compare individual species’ abundances between bladder cancer and normal samples. A total of 35 species were found to be differentially abundant between these disease states (*p* < 0.05) (Figure 3A). Most notably, *Caldanaerobacter subterraneus* was significantly enriched in the urine samples of patients with bladder cancer. Among others, the abundance of *Burkholderia ambifaria*, *Staphylococcus xylosus*, and *Klebsiella pneumoniae* was significantly lesser in these samples. The significance (Figure 3B) and fold-change (Figure 3C) distributions of species were plotted. A phylogenic tree was created to visualize the distribution of differentially abundant species by class or phylum (Figure 3D). A majority of the differentially abundant species were of the classes Betaproteobacteria and Gammaproteobacteria.

### 3.4. Microbe-Associated Immune Dysregulation

Gene set enrichment analysis was conducted to determine whether select immune-associated pathways were enriched with respect to individual species’ abundances. Twenty-two KEGG immune pathways were assessed for enrichment with each of the 35 differentially abundant species above (see Section 2). Nominal enrichment scores (NESs) and test statistics were calculated for each species–pathway combination. Several immune pathways show significant enrichment with respect to one or more of these species (Figure 4A), including antigen processing and presentation, cytosolic DNA sensing, and leukocyte transendothelial migration. Notably, samples with greater abundances of *Pseudomonas fluorescens* and *Pseudomonas putida* corresponded to the positive enrichment of these pathways. These species were less abundant in bladder cancer samples, with lesser abundance correlating to the downregulation of these pathways.

Spearman’s correlations were further computed to assess these differentially abundant species for their implications in cytokine dysregulation. Correlation coefficients and test statistics were calculated of each species–cytokine combination. *Pseudomonas fluorescens* and *Pseudomonas aeruginosa* were significantly positively correlated to these cytokines’ expressions, particularly for CCL3L1, IL1A, and IL4R (Figure 4B). Among others, numerous significant correlations were also observed of the species *Cutibacterium acnes*, *Cupriavidus metallidurans*, and *Stenotrophomonas maltophilia* to the expression of CCL3L1, IL1A, IL1B, IL1RAP, and IL4R.

### 3.5. Microbe-Associated Adherens Junction and WNT Signaling Enrichment

Due to their implications in the transition from non-muscle-invasive to muscle-invasive bladder cancer, EMT and the WNT signaling pathway were assessed for enrichment with respect to each of the 35 differentially abundant species above. The adherens junction KEGG pathway is composed of genes involved in intercellular adhesion interactions. It is closely associated with the genes involved in EMT, and thus was chosen to model this pathway [45]. Enrichment plots were created to visualize the running enrichment score of the 10 differentially abundant species of the greatest significance (Figure 5A). These pathways show positive enrichment with respect to a majority of these species, including *Pseudomonas fluorescens* and *Pseudomonas putida*. These species were less abundant in bladder cancer samples, with lesser abundance correlating to the downregulation of these pathways. Spearman’s correlations were computed to further assess differentially abundant species for correlation to the expression of each of the genes in these pathways. Correlation coefficients and test statistics were calculated for each species–gene combination. Numerous species were of significant correlation to these genes’ expression, with a vast majority of these correlations being positive (Figure 5B). The greater abundance of the species *Cupriavidus metallidurans* correlated to the increased expression of many of these genes, including ACTN1, CSNK2A1, NECTIN1, PTPRF, and RAC3 of the adherens junction pathway and CCND2, C2NKA1, FZD5, LRP6, RAC2, and RAC3 of the WNT signaling pathway.

### 3.6. Validation of Differential Abundance

We further attempted to validate whether the differentially abundant species identified above were of similar significance in a fourth dataset. After performing contamination correction (see Section 2), the Kruskal–Wallis test was again used to identify species that were of significantly altered abundance between the bladder cancer and normal urine samples. In total, 12 species were differentially abundant between these disease states (*p* < 0.05) (Figure 6A). Hypergeometric testing was used to quantify the extent of overlap between these 12 species and the 35 identified in the original datasets. At the genus-level, seven of these features were common to both datasets (Figure 6B). This yielded a test statistic of 7.86 × 10^−7^, indicating a significant amount of overlap between the datasets. We present the following common genera for their potential as microbial urinary biomarkers of bladder cancer: *Escherichia*, *Acinetobacter*, and *Enterobacter*. Species of the genera *Escherichia* and *Acinetobacter* were consistently of lesser abundance in bladder cancer samples, and species of the genus *Enterobacter* were of greater abundance in bladder cancer samples.

## 4. Discussion

At a global level, bacterial abundance was found to vary significantly between bladder cancer and normal urine samples. Absolute and relative diversity values were generally lesser in the urine samples of patients with bladder cancer. A similar decrease in microbial diversity with bladder cancer has been reported previously, though studies lack consistency [46,47,48]. Through PCoA, we observed a global variation in microbiome composition between the urine samples of patients with bladder cancer and those of healthy individuals. Dysbiosis of the gut microbiome is known to be implicated in numerous human diseases [27,28], largely through differential immune modulation and metabolic interactions [29,30]. However, the extent to which this is true of the urinary microbiome is unclear. Regardless of the causal or resultant nature of this dysbiosis, the observed variations in the urinary microbiome may be highly useful in the diagnosis of bladder cancer. The detection of a urinary microbial profile distinct for patients with bladder cancer may provide an accurate, non-invasive means of diagnosis.

At a species level, we observed the differential abundance of specific taxa in bladder cancer urine samples. In total, 35 species were found to be differentially abundant between the samples’ urinary microbiomes. A majority of these species were of the phyla Proteobacteria, Actinobacteria, Firmicutes, and Cyanobacteria. The differential abundance of these phyla has been identified by other studies of the urinary microbiome in bladder cancer [46,47,48]. At a genera level, we observed several consistencies in differential abundance, as well. Numerous species of the genus *Escherichia* were of lesser abundance in the bladder cancer samples. We also discovered a consistency in the enrichment of *Cupriavidus* in bladder cancer samples [47], and a decreased abundance of the genus *Acinetobacter*. After validation, we ultimately discovered that species of the following genera were consistently differentially abundant between bladder cancer and normal samples: *Escherichia*, *Acinetobacter*, and *Enterobacter*. Through further investigation, we may determine whether these genera and others are clinically relevant. We suspect that these genera may prove highly useful as biomarkers for this disease. Additional research on a greater number of patients is needed to confirm the implications of these genera and to determine their potential for use in a noninvasive diagnostic approach.

Alternatively, these findings might suggest the utility of microbial-based cancer therapies. The gut microbiome has been shown to influence the efficacy of many cancer therapies, including immunotherapy, chemotherapy, radiation therapy, and even surgery [49]. Studies have demonstrated the ability of fecal transplants and dietary interventions to increase a patient’s likelihood of responding to anti-cancer treatments [50]. These interventions are designed to modulate the gut microbiome, and less is known regarding how they might affect the urinary microbiome. Nonetheless, these approaches prove highly promising. Given the relevance of the urinary microbiome as demonstrated above, additional research into microbial-based cancer therapies may similarly be of use for bladder cancers.

We further discovered the enrichment of specific immune pathways with respect to the differentially abundant species identified. Among others, *Pseudomonas fluorescens* and *Pseudomonas putida* consistently correlated to the enrichment of antigen processing and presentation pathways. These species were of decreased abundance in bladder cancer samples, corresponding to decreased antigen processing and presentation. Cancer cells are known to exhibit this trait as means of immune evasion, by which a reduction in antigen presentation prevents the activation of a host’s immune response [51,52]. In this way, the reduced abundance of these species may contribute to a cancer’s evasion of immune recognition. Similarly, cytosolic DNA sensing pathways were enriched with these species. A reduction in their abundance values in bladder cancer samples corresponded to a reduction in DNA sensing. The suppression of this pathway is also a known mechanism of immune evasion exhibited by cancer cells [53,54]. The lesser abundance of *Pseudomonas fluorescens* and *Pseudomonas putida*, as observed in bladder cancer samples, may provide cancer cells a greater means of evading immune recognition, ultimately promoting tumorigenesis. The microbial-associated enrichment of these pathways may act as a mechanistic link between the human microbiome and the pathogenesis of bladder cancer. Further investigation of these taxa and their specific metabolic interactions may prove useful toward understanding their implications in this disease.

As expected, these species were also found to correlate to differential cytokine expression. Among others, numerous significant correlations were observed between the species *Cutibacterium acnes*, *Cupriavidus metallidurans*, and *Stenotrophomonas maltophilia* and the cytokines CCL3L1, IL1A, IL1B, IL1RAP, and IL4R. Cytokine dysregulation is known to alter a local immune landscape through the differential recruitment of immune cells and the promotion of an inflammatory response [55]. The dysregulation of cytokine activity is heavily implicated in oncogenesis [55] and may mediate the effect of the urinary microbiome on bladder cancer development or progression. Through the immune modulatory relations demonstrated, the urinary microbiome may be implicated in the pathology of bladder cancers, with dysbiosis ultimately promoting a bladder cancer’s development and progression.

EMT has been shown to be characteristic of the transition from non-muscle-invasive to muscle-invasive bladder cancer [17,18,19]. The genes involved in EMT are heavily associated with a decrease in cell adhesion, with many of them also being integral to the adherens junction pathway [20]. The destabilization of the adherens junction pathway is associated with increased cellular motility and metastasis resultant of decreased intercellular adhesion [56,57]. We observed the enrichment of the EMT pathway with respect to several microbial species. The decreased abundance of *Pseudomonas fluorescens* and *Pseudomonas putida* corresponded to decreased pathway activity, and thus decreased intercellular adhesion in bladder cancer samples. The observed decrease in these species’ abundances may suggest their relation to EMT and to the pathology of bladder cancer as a whole. The adherens junction pathway has been previously implicated in many other cancer types, including colorectal cancer, breast cancer, and lung cancer [58,59,60,61,62]. Moreover, dysbiosis of the gut microbiome has also been shown to contribute to epithelial dysregulation [63]. We propose that the urinary microbiome may play a similar role in this pathway’s regulation within the bladder. Functional metabolic analyses may further elucidate the urinary microbiome’s implications in the adherens junction pathway and EMT.

The WNT pathway, too, has been shown to play an oncogenic role in bladder cancers [21,22]. It is thought to be involved in the transition from non-muscle-invasive to muscle-invasive bladder cancer and has been used as an important diagnostic and prognostic biomarker [21,22]. This pathway plays a central role in regulating cell differentiation, providing a suitable explanation as to its implications in cancer growth and metastasis [64]. We also observed the consistent upregulation of the WNT pathway with respect to several species’ abundances. *Pseudomonas fluorescens* and *Pseudomonas putida* were of decreased abundance in bladder cancer samples, corresponding to the decreased activity of this pathway. These species may contribute to this pathway’s dysregulation, ultimately yielding the decreased regulation of cell differentiation in bladder cancer samples. Various species have been implicated in the molecular alteration of the WNT glycoproteins, ultimately destabilizing its components [65]. Bacterial virulence factors, too, have been shown to modulate this pathway through an array of mechanisms, including the repression of WNT inhibitors, the blocking of WNT–Fzd ligands, and the dysregulation of WNT ligands’ expressions [65]. We propose that the urinary microbiomes may dysregulate this pathway through similar means, consequently promoting cancer development and progression. Further metabolic analyses may confirm these hypotheses.

Moreover, the microbial-associated dysregulation of the adherens junction and WNT signaling pathways may mutually influence one another. Many genes are common between these pathways, perhaps explaining the observed similarities in their dysregulation. Genes of the WNT signaling pathway are known to interact with the β-catenin protein of the adherens junction cascades, resulting in the detachment of E-cadherin from a cell’s cytoskeleton [66]. In this way, the microbial-associated dysregulation of these pathways may cyclically influence one another, driving a cancer toward the muscle-invasive subtype.

## 5. Conclusions

Ultimately, we have observed significant dysbiosis in the urinary microbiome of patients with bladder cancer. We discovered microbial-associated enrichment of immune pathways and cytokine activity, suggesting the potential immune modulatory relation of these species. The dysregulation of EMT and WNT/β-catenin signaling was also observed with these species, suggesting that the observed decreases in their abundances in bladder cancer samples may reflect the transition from the non-muscle-invasive to the muscle-invasive subtype. With these considerations, we may better understand the urinary microbiome for its implications in bladder cancer and for its implications in the acquisition of the muscle-invasive subtype. By further exploring the species that relate to this acquisition, we may be more equipped to develop urinary biomarker tests that are more capable of detecting less-advanced bladder cancers. Ultimately, the creation of a panel of urinary biomarker species may prove highly useful toward the non-invasive diagnosis of bladder cancer, ideally providing precision regardless of a cancer’s stage or subtype.

In summary, our results suggest that the urinary microbiome may provide a non-invasive diagnostic tool of significant sensitivity and specificity, allowing for the earlier diagnosis of bladder cancer. Our results are limited due to the correlational nature of this study. We are unable to claim a causal relationship between dysbiosis of the urinary microbiome with bladder cancer pathogenesis. Due to the limited number of samples investigated, these results are further limited. Additional samples are necessary to confirm whether the differentially abundant species identified are clinically relevant, as is a greater variation in the disease progression of patients. Moreover, we acknowledge that potentially confounding variables might be present among these samples, including differences in the patients’ types of bladder cancer, the patients’ subtypes, the patients’ genders, and the urine collection methods used [67]. This information was not available for each of the chosen datasets, preventing us from further stratifying our results. The sample collection and sequencing procedures used by these studies also inherently differed. We attempted to mitigate these differences through the above normalization procedures, though the potentially confounding effects should be noted. Lastly, species-level profiling performed with the use of a reference sequence database will only capture culturable species and may omit species otherwise present. This is common in most microbiome studies that employ direct sequence alignment.

## Figures and Tables

**Figure 1 cancers-16-00394-f001:**
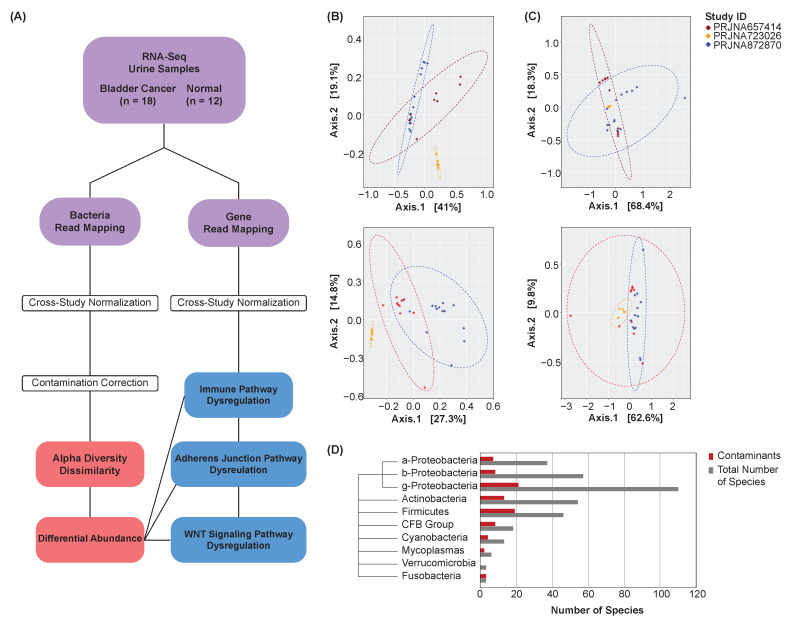
Analysis overview. (**A**) Overview schematic. (**B**) PCoA plots of unnormalized bacterial abundance (top) and gene expression (bottom) profiles by study. Points represent samples, with a closer proximity indicating greater similarity in the samples’ bacterial abundance or gene expression profiles. (**C**) PCoA plots of normalized bacterial abundance (top) and gene expression (bottom) profiles by study. MAD normalization was performed. Points represent samples, with a closer proximity indicating greater similarity in the samples’ bacterial abundance or gene expression profiles. Samples show greater comparability after normalization. (**D**) Phylogenic tree and bar chart of contaminant species by class or phylum.

**Figure 2 cancers-16-00394-f002:**
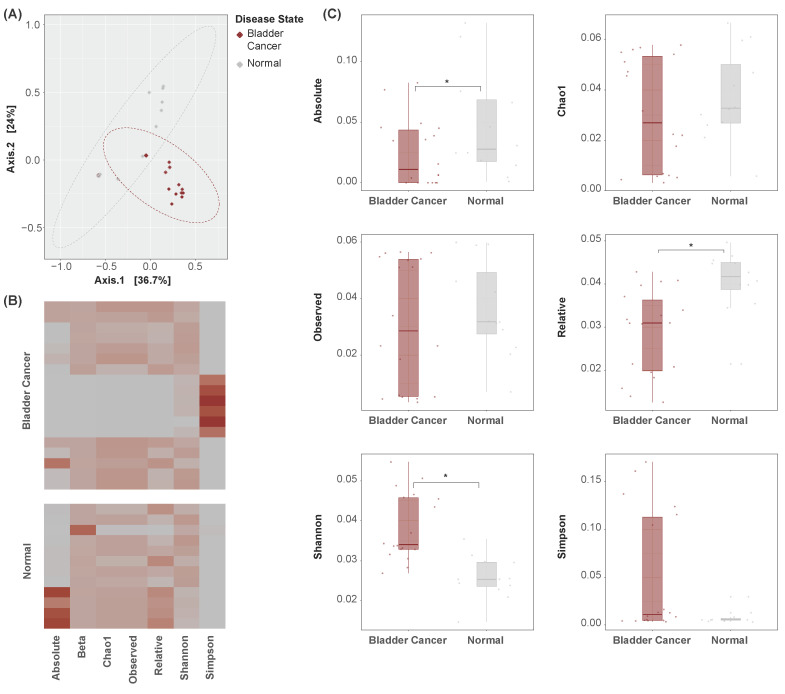
Global indicator dissimilarity. (**A**) PCoA plots of bacterial abundance by disease state. Points represent samples, with a closer proximity indicating greater similarity in the samples’ bacterial abundance profiles. Bladder cancer samples occupy a distinct region of the plot as compared to normal samples, suggesting global dissimilarity in their urinary microbiome compositions. (**B**) Heatmap of alpha and beta diversity measures across samples. Rows represent samples. Cell values are relative, and units are arbitrary. (**C**) Boxplots of alpha diversity measures by disease state. Absolute, relative, and Shannon diversity indicators are significantly altered between bladder cancer and normal samples. * *p* < 0.05.

**Figure 3 cancers-16-00394-f003:**
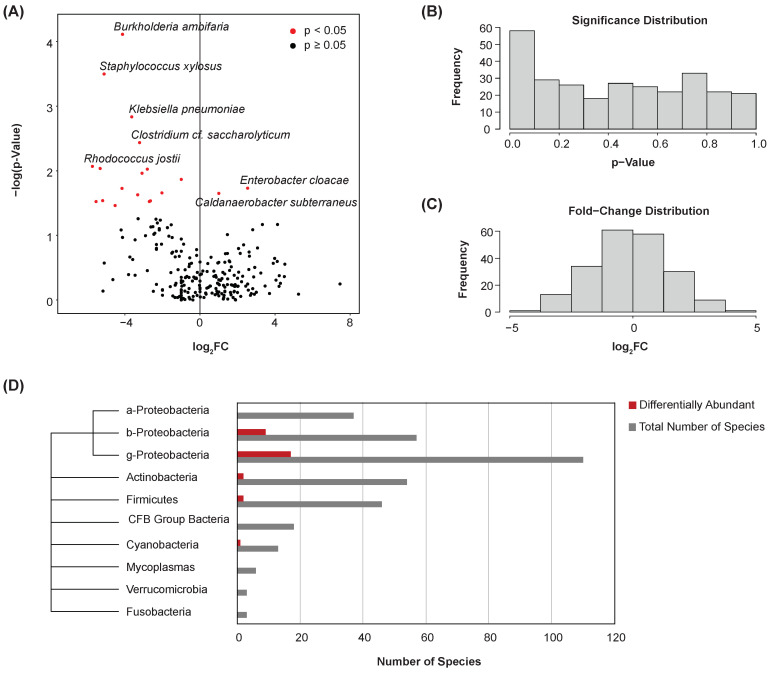
Differential abundance. (**A**) Volcano plot showing differential abundance of species between bladder cancer and normal samples. Points represent microbes. Thirty-five species were determined to be differentially abundant. (**B**) Histogram showing the significance distribution of species. (**C**) Histogram showing the fold-change distribution of species. (**D**) Phylogenic tree and bar chart of differentially abundant species by class or phylum.

**Figure 4 cancers-16-00394-f004:**
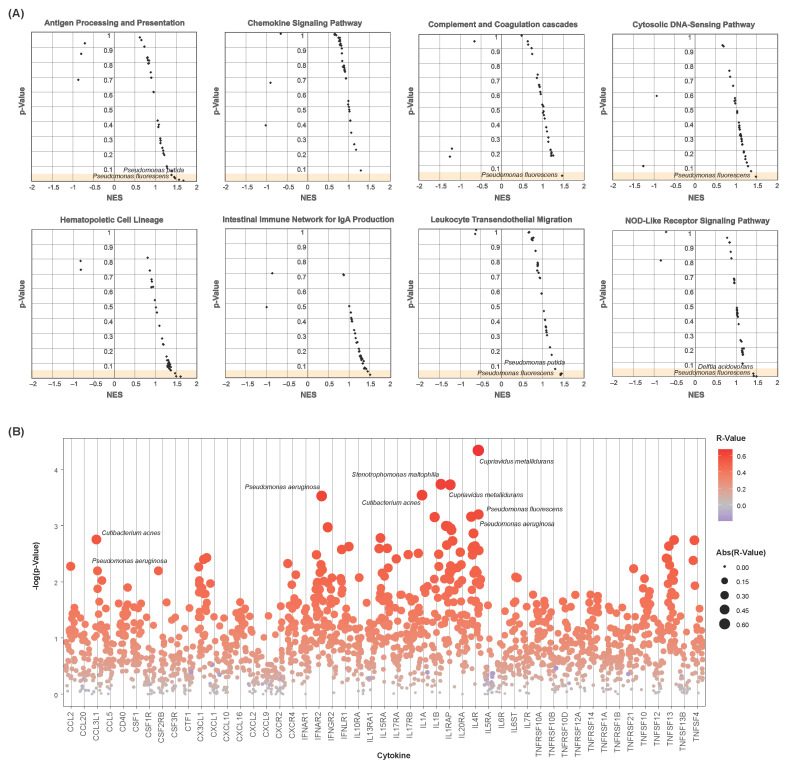
Microbial-associated immune dysregulation. (**A**) Scatter plots showing microbial-associated enrichment of select immune pathways. Nominal enrichment scores (NESs) and *p*-values are plotted. Each point represents a microbe–pathway combination. Only differentially abundant microbes were chosen for analysis. (**B**) Strip chart of microbe–cytokine correlations. *p*-values are plotted. Each point represents a microbe–cytokine combination, with sizes and colors indicating their correlation coefficients (R-value). Only differentially abundant microbes were chosen for analysis. Cytokines show positive correlation to a vast majority of the differentially abundant species.

**Figure 5 cancers-16-00394-f005:**
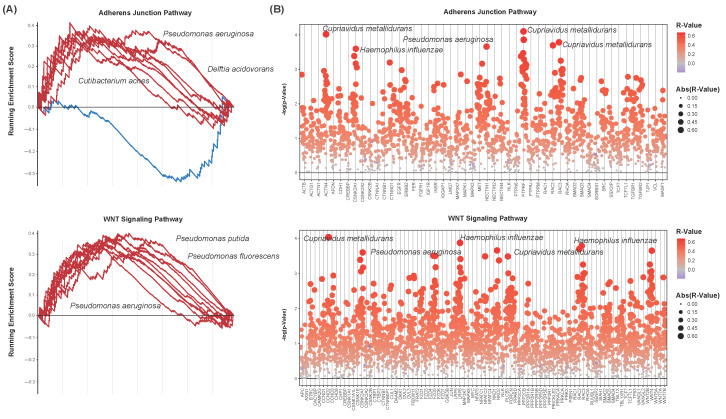
Microbial-associated adherens junction and signaling pathway dysregulation. (**A**) Enrichment plots showing the running enrichment score of select microbes to the adherens junction (top) and WNT signaling (bottom) pathways. Lines represent microbes. Only the ten microbes of the greatest significance in differential abundance are shown. Both pathways are positively enriched with respect to a majority of the microbes analyzed. (**B**) Strip charts of microbe–gene set correlations. Genes of the adherens junction (top) and WNT signaling (bottom) pathways are shown. *p*-values are plotted. Each point represents a microbe–gene combination, with sizes and colors indicating their correlation coefficients (R-value). Only differentially abundant microbes were chosen for analysis. Gene sets show positive correlation to a vast majority of the differentially abundant species.

**Figure 6 cancers-16-00394-f006:**
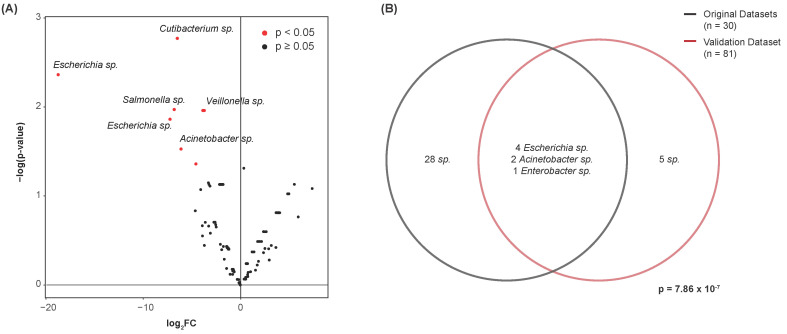
Validation of differentially abundant species by genus. (**A**) Volcano plot showing differential abundance of species between bladder cancer and normal samples. Points represent microbes. Twelve species were determined to be differentially abundant. (**B**) Venn diagram showing the number of differentially abundant species common to both the original datasets and the validation dataset. Species of the genera *Escherichia*, *Acinetobacter*, and *Enterobacter* were commonly differentially abundant. Hypergeometric testing revealed a significant amount of overlap between these datasets, with 35 species identified in the original datasets, 12 species identified in the validation dataset, and 602 noncontaminant species present in both datasets collectively.

## Data Availability

The data can be accessed through the NCBI BioProject Database under the accessions PRJNA657414, PRJNA872870, and PRJNA723026, and the Genome Sequencing Archive under the accession PRJCA003781.

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
