# Peer review of "Urinary Microbiome Dysbiosis and Immune Dysregulations as Potential Diagnostic Indicators of Bladder Cancer"

_cancers, 2024, doi:10.3390/cancers16020394_

Round 1

Reviewer 1 Report

Comments and Suggestions for Authors

<Summary>

This study investigated to characterize bladder cancer-specific dysbiosis of the urinary microbiome to explore its diagnostic potential. Interestingly, the study suggests that the urinary microbiome may reflect dysregulations of the tumor microenvironment. However, the clinical utility is not clear, so quality of this manuscript is not enough to be accepted in this journal as it stands. Specifically, the sample size was very small (only 18 and 12) and there was significant lack of validation.

1. Figure 1

Figure 1B has weird characters. These should be modified.

2. Methods

Urine samples are collected by voiding or catheterization? Collection method should be consistent between cancers and controls. Voiding urine is contaminated with urethral microbiome. This can affect the results.

3. Discussion

The authors should come up with interventions, such as improving outcomes. Do the authors think that anti-microbiome drug can change the prognosis or sensitivity to anti-cancer therapy? These topics should be added to the Discussion section.

4. General

This study is just on start-line of clinical application of urine microbiome. The results regarding dysbiosis of urine samples should be validated in the independent cohort.

Author Response

This study investigated to characterize bladder cancer-specific dysbiosis of the urinary microbiome to explore its diagnostic potential. Interestingly, the study suggests that the urinary microbiome may reflect dysregulations of the tumor microenvironment. However, the clinical utility is not clear, so quality of this manuscript is not enough to be accepted in this journal as it stands. Specifically, the sample size was very small (only 18 and 12) and there was significant lack of validation.

Dear reviewer,

We thank you for your comments and consideration throughout this process. We acknowledge the small sample size and have further highlighted this limitation in the discussion section. We have attempted to validate our findings using a fourth dataset, and have included an additional figure to summarize our findings. We hope the following revisions and others significantly augment the validity of this study.

  1. Figure 1

Figure 1B has weird characters. These should be modified.

We do not see these characters ourselves, but we suspect there may have been an issue in the process of uploading the files. We will work with the editorial office to ensure that this figure is correct if we proceed with publication.

  1. Methods

Urine samples are collected by voiding or catheterization? Collection method should be consistent between cancers and controls. Voiding urine is contaminated with urethral microbiome. This can affect the results.

Unfortunately, the included studies do not distinguish between these collection methods. We have included this limitation and an appropriate reference in the discussion section.

  1. Discussion

The authors should come up with interventions, such as improving outcomes. Do the authors think that anti-microbiome drug can change the prognosis or sensitivity to anti-cancer therapy? These topics should be added to the Discussion section.

We have included additional commentary on these and related therapies in the discussion section.

  1. General 

This study is just on start-line of clinical application of urine microbiome. The results regarding dysbiosis of urine samples should be validated in the independent cohort.

We have attempted to validate our results for differential abundance using a fourth study with 62 bladder cancer samples and 19 healthy samples. We’ve included an additional figure to summarize our findings.

Reviewer 2 Report

Comments and Suggestions for Authors

This study evaluates the urinary microbiome disbyosis of bladder cancer patients, its potential role as a diagnostic biomarker of bladder cancer, and its association with  differential cytolkine expression.  The study shows significant novelty, is well performed and is clearly written.

Comments:

·         I think that the manuscript clearly needs a paragraph including the limitations of the study, quite specially:

o   The low sample size, specially significant when using high-throughput techniques; and considering that urine samples are quite accesible.

o   The lack of a causal relationship between the different observations (this is mentioned in the last paragraph, but I think that it should be placed in context with the sample size).

·         I think that the authors should comment if the use of RNA-Seq data from three different studies, which have possible been sequenced at different moments -rather than in one single process - may have an impact on the sequencing results.

·         The langauge of the manuscript should be adapted to the lack of causality. For example, in line 241 “These species were less abundant in bladder cancer samples, meaning these pathways, too, were downregulated.” seems to impliy that there is an actual functional association between the different observations.

Author Response

This study evaluates the urinary microbiome disbyosis of bladder cancer patients, its potential role as a diagnostic biomarker of bladder cancer, and its association with differential cytolkine expression.  The study shows significant novelty, is well performed and is clearly written.

Dear reviewer,

We thank you for your comments and consideration throughout this process. We hope the following revisions and others significantly augment the validity of this study.

I think that the manuscript clearly needs a paragraph including the limitations of the study, quite specially:

The low sample size, especially significant when using high-throughput techniques; and considering that urine samples are quite accesible.

The lack of a causal relationship between the different observations (this is mentioned in the last paragraph, but I think that it should be placed in context with the sample size).

We acknowledge the small sample size and have further highlighted this limitation in the discussion section. We have added additional commentary relating this limitation to the correlational nature of the study. We have attempted to validate our findings using a fourth dataset, and have included an additional figure to summarize our findings.

I think that the authors should comment if the use of RNA-Seq data from three different studies, which have possible been sequenced at different moments -rather than in one single process - may have an impact on the sequencing results.

We hope that the normalization procedures serve to minimize these and other related batch effects. We’ve included additional commentary of this limitation in the discussion section.

The langauge of the manuscript should be adapted to the lack of causality. For example, in line 241 “These species were less abundant in bladder cancer samples, meaning these pathways, too, were downregulated.” seems to impliy that there is an actual functional association between the different observations.

We’ve attempted to identify and replace all clauses of this nature to ensure that our findings are being accurately conveyed.

Reviewer 3 Report

Comments and Suggestions for Authors

Overall impression:

There is a paucity of data on the microbiome in bladder cancer, and the authors successfully add meaningful work to the existing literature. The analysis of immune pathways that may be dysregulated in association with alterations in the urinary microbiome is novel and important.

It's not clear if the most important message is that the urinary microbiome could be a diagnostic tool (as suggested by the title and concluding paragraph). There are already similar small studies looking at the microbiome in bladder cancer vs. healthy controls, many with heterogeneous results that need to be further examined in large cohorts. Therefore, the authors may consider emphasizing the novelty of their work, and that understanding how the microbiome may influence established mechanisms of oncogenesis in bladder cancer could have important implications for prevention and therapeutics.

Title

·       I would suggest incorporating mention of the dysregulation of immune pathways in the title. This may help with readership later; there are already other articles about urinary microbiome as a biomarker, but there aren’t any that look how the urinary microbiome may drive immune pathway dysregulation.

Abstract

·    Would consider adding mention of the number of patients and controls along with the type of cancer they had (muscle or non-muscle invasive).

Introduction

·         Would make the introduction more succinct.

·         Again, I’m not sure the data presented here make a strong argument for using the microbiome as a diagnostic tool because it lacks patient data—therefore this may not need to be emphasized in the introduction.

·         Line 99 – this is the incorrect citation for this sentence. The authors’ prior work with EMT in MIBC is reference 38 but it is listed here as 39. Authors should go through each citation and ensure numbers match correct reference.

Methods

·         Line 140-143 of the methods is repeated in results (174-179). Would remove from one or the other.

·         Is patient data available regarding the type of bladder cancer? Were these all muscle-invasive cancers? All urothelial? Males/females? Any data describing when or how the urine was collected for bladder cancer patients? If available, this data should be included as it would strengthen the paper.

Results

·         Figure 1D. This figure includes both bladder cancer and health controls combined, correct? If so, authors could clarify that.

·         Figure 3: Some of the font/wording came through as symbols on the PDF rather than words (for example, the X-axis label on 3D). Would confirm these are clear/correct.

·         Figure 3D: Is it possible to compare more clearly healthy controls vs. bladder cancer in this plot?

·         Figure 5B: the vertical lines on the figure make it difficult to read the words of the bacteria. Could make the lines very light gray or remove.

Discussion

·         It is worth noting in the discussion that since bladder cancer patient data was taken from multiple studies, there could have been significant variation in specimen collection and processing that would impact the results.

·         The logic/reasoning behind the argument in lines 392-400 could be clarified.

References

·         Ensure references are properly numbered.

Author Response

Overall impression:

There is a paucity of data on the microbiome in bladder cancer, and the authors successfully add meaningful work to the existing literature. The analysis of immune pathways that may be dysregulated in association with alterations in the urinary microbiome is novel and important.

It's not clear if the most important message is that the urinary microbiome could be a diagnostic tool (as suggested by the title and concluding paragraph). There are already similar small studies looking at the microbiome in bladder cancer vs. healthy controls, many with heterogeneous results that need to be further examined in large cohorts. Therefore, the authors may consider emphasizing the novelty of their work, and that understanding how the microbiome may influence established mechanisms of oncogenesis in bladder cancer could have important implications for prevention and therapeutics.

Dear reviewer,

We thank you for your comments and consideration throughout this process. We acknowledge the small sample size and have further highlighted this limitation in the discussion section. We have attempted to validate our findings using a fourth dataset, and have included an additional figure to summarize our findings. We have attempted to further highlight the discovered pathway dysregulations in the title and main text. We hope the following revisions and others significantly augment the validity of this study.

Title

I would suggest incorporating mention of the dysregulation of immune pathways in the title. This may help with readership later; there are already other articles about urinary microbiome as a biomarker, but there aren’t any that look how the urinary microbiome may drive immune pathway dysregulation.

We agree that the discovered pathway dysregulations are of great novelty in this study. We have accordingly changed the title to “Urinary Microbiome Dysbiosis and Immune Dysregulations as Potential Diagnostic Indicators of Bladder Cancer”.

Abstract

Would consider adding mention of the number of patients and controls along with the type of cancer they had (muscle or non-muscle invasive).

We have included the sample sizes in the abstract. Unfortunately, the studies analyzed do not all mention their samples’ subtypes. We have included this limitation in the discussion section.

Introduction

Would make the introduction more succinct.

Again, I’m not sure the data presented here make a strong argument for using the microbiome as a diagnostic tool because it lacks patient data—therefore this may not need to be emphasized in the introduction.

We have attempted to validate our findings using a fourth dataset, and have included an additional figure to summarize our findings. We hope that this data further supports the utility of the urinary microbiome as a diagnostic tool.

Line 99 – this is the incorrect citation for this sentence. The authors’ prior work with EMT in MIBC is reference 38 but it is listed here as 39. Authors should go through each citation and ensure numbers match correct reference.

We believe that both the in-text citation and the full citation read 39. We have worked to ensure that all citations are correct.

Methods

Line 140-143 of the methods is repeated in results (174-179). Would remove from one or the other.

We have removed these lines from the results sections.

Is patient data available regarding the type of bladder cancer? Were these all muscle-invasive cancers? All urothelial? Males/females? Any data describing when or how the urine was collected for bladder cancer patients? If available, this data should be included as it would strengthen the paper.

As mentioned above, the studies analyzed unfortunately do not all mention their samples’ subtypes. They also do not all distinguish between urothelial and other carcinomas, or provide information on the collection methods. Both male and female patients are included. We have included these limitations in the discussion section.

Results

Figure 1D. This figure includes both bladder cancer and health controls combined, correct? If so, authors could clarify that.

That is correct. Contamination correction was performed on all samples collectively. We have included this information in the results section.

Figure 3: Some of the font/wording came through as symbols on the PDF rather than words (for example, the X-axis label on 3D). Would confirm these are clear/correct.

We do not see these characters ourselves, but we suspect there may have been an issue in the process of uploading the files. We will work with the editorial office to ensure that this figure is correct if we proceed with publication.

Figure 3D: Is it possible to compare more clearly healthy controls vs. bladder cancer in this plot?

This plot is showing the phylogenic division of species that were found to be differentially abundant between bladder cancer and normal samples. Due to the nature of this comparison, no species can be isolated to either group.

Figure 5B: the vertical lines on the figure make it difficult to read the words of the bacteria. Could make the lines very light gray or remove.

We’ve decreased the weights of these lines and of those in Figure 4B.

Discussion

It is worth noting in the discussion that since bladder cancer patient data was taken from multiple studies, there could have been significant variation in specimen collection and processing that would impact the results.

We hope that the normalization procedures serve to minimize these and other related batch effects. We’ve included additional commentary of this limitation in the discussion section.

The logic/reasoning behind the argument in lines 392-400 could be clarified.

We have worked to clarify this conclusion paragraph.

References

Ensure references are properly numbered.

We have worked to ensure that all citations are correct.

Round 2

Reviewer 1 Report

Comments and Suggestions for Authors

The authors have modified the mansucript properly.